# Choline, Neurological Development and Brain Function: A Systematic Review Focusing on the First 1000 Days

**DOI:** 10.3390/nu12061731

**Published:** 2020-06-10

**Authors:** Emma Derbyshire, Rima Obeid

**Affiliations:** 1Nutritional Insight, Surrey KT17 2AA, UK; 2Department of Clinical Chemistry, University Hospital of the Saarland, Building 57, 66424 Homburg, Germany; rima.obeid@uks.eu

**Keywords:** brain function, choline, early life, first 1000 days of life, lactation, neurological development, pregnancy

## Abstract

The foundations of neurodevelopment across an individual’s lifespan are established in the first 1000 days of life (2 years). During this period an adequate supply of nutrients are essential for proper neurodevelopment and lifelong brain function. Of these, evidence for choline has been building but has not been widely collated using systematic approaches. Therefore, a systematic review was performed to identify the animal and human studies looking at inter-relationships between choline, neurological development, and brain function during the first 1000 days of life. The database PubMed was used, and reference lists were searched. In total, 813 publications were subject to the title/abstract review, and 38 animal and 16 human studies were included after evaluation. Findings suggest that supplementing the maternal or child’s diet with choline over the first 1000 days of life could subsequently: (1) support normal brain development (animal and human evidence), (2) protect against neural and metabolic insults, particularly when the fetus is exposed to alcohol (animal and human evidence), and (3) improve neural and cognitive functioning (animal evidence). Overall, most offspring would benefit from increased choline supply during the first 1000 days of life, particularly in relation to helping facilitate normal brain development. Health policies and guidelines should consider re-evaluation to help communicate and impart potential choline benefits through diet and/or supplementation approaches across this critical life stage.

## 1. Introduction

The first 1000 days (2 years) are life are fundamental as the foundations of neurodevelopment are laid down and an adequate supply of nutrients, particularly folic acid, iron, iodine, and choline, amongst others are crucial in achieving this [1,2]. Choline is an ‘essential’ nutrient, and humans can produce choline in small amounts through the hepatic phosphatidylethanolamine N-methyltransferase (PEMT) pathway. However, most individuals need to consume choline from dietary sources to augment the choline that is produced endogenously and prevent deficiency [3,4]. The fetus and neonate have particularly high choline demands that participate in normal development [5]. Over the last few million years, the hominin brain has tripled in size and has vast nutritional needs, which includes a suitable choline supply [6,7,8]. Inadequate choline supplies could potentially perturb fetal progenitor cell migration, proliferation, apoptosis, and differentiation, thereby adversely modifying brain development [8]. Impairments resulting from choline deficits can be irreversible [9]. Subsequently, failure to provide choline during the first 1000 days could result in lifelong deficits in brain function [9].

### 1.1. Functions

Chemically, choline is closely associated with the B-vitamin family [10]. It is an important starter material facilitating the biosynthesis of a metabolite that plays central roles in fetal development, especially the brain [11]. Metabolically, choline is involved in acetylcholine, phospholipid, and betaine synthesis [12]. Additionally, choline is a precursor of the gut bacteria-derived metabolites, trimethylamine N-oxide [13]. Betaine is a bioactive compound that can spare some of choline’s actions and is found in wheat bran, wheat germ, and spinach, thereby making it a useful food for pregnant women who are vegetarian or vegan [14,15,16].

Phosphatidylcholine (PtdCho) is a major component of cellular membranes and is needed for cell division and growth—particularly that of the fetal brain, which occurs rapidly during the sixth month of pregnancy and continues up until about 3–5 years of age [11,17]. PtdCho plays a role in cell signaling as a PtdCho donor to synthesize sphingomyelin from ceramide and is the predominant carrier of arachidonic and docosahexaenoic acid in plasma [18]. Sphingomyelin—a PtdCho derived phospholipid, also referred to as sphingolipids, are characteristic of nervous tissue and are required for the myelination of nerve fibers (axons) in both the central and peripheral nervous systems [11,19]. Sphingomyelin also produces cellular signaling molecules, including ceramide, diacylglycerol, and/or platelet-activating factor, all of which have reputed roles in fetal development [11,20].

In physiologic terms, choline is important for lipid metabolism and brain, liver, and muscle function [9]. Metabolically choline plays a key role in the transport and metabolism of lipids and cholesterol. PtdCho comprises around 70–95% phospholipids in lipoproteins, and about 50% of these are choline [21,22]. Choline is a central part of phospholipids which are abundant in all biological membranes, where approximately 40–50% of cellular membrane phospholipids are comprised of PtdCho. They play central roles in the function and structure of membranes, including their signaling, transport, and repair [9,21,22]. It has long been established that choline is required for brain neurotransmitter acetylcholine synthesis, which is dependent on dietary choline consumption when the body’s do novo synthesis becomes insufficient [23,24,25]. Fetal progenitor cells which differentiate, migrate, proliferate, and undergo apoptosis also require choline [8].

It should be recognized that single nucleotide polymorphisms (SNP) in choline metabolizing genes can alter: (1) the use of choline as a methyl donor, (2) the partitioning of dietary choline between betaine and phosphatidylcholine synthesis, and (3) the distribution of dietary choline between cytidine diphosphate-choline and PEMT pathways, which could influence the choline requirements of pregnant women [26,27]. Mechanistically, other work [28] shows enhanced use of choline for phosphatidylcholine production via both the cytidine diphosphate-choline and PEMT pathways, further indicating substantial demands for choline in late pregnancy.

Choline can modify gene methylation and expression, and thereby alter neuronal activity [29]. Data from animal models show that higher intakes of choline during gestation and the perinatal period can protect against brain damage and cognitive/neurological declines associated with epilepsy and certain hereditary conditions, such as Down Syndrome and Rett Syndromes [30]. Additionally, a review collating evidence from 22 human clinical and preclinical studies concluded that choline supplementation has the potential to ameliorate some of the behavioral, cognitive, and neurological deficits observed in offspring exposed to alcohol in utero [31].

### 1.2. Choline, the Brain, and Neurons

In early life “brain development relies on complex and intermingled mechanisms during gestation and post-natal months, with intense interactions between genetic, epigenetic and environmental factors” [32]. Of these mechanisms, choline is regarded to have pivotal roles in the structural integrity of membranes, neurotransmission, and methyl group metabolism [33].

Choline passes through the blood-brain barrier by facilitated diffusion governed by the choline concentration gradient, and it is stored in the brain as membrane-bound phospholipids that are hydrolyzed by choline acetyltransferase to yield choline for acetylcholine synthesis [21,34]. It has been proposed that brain phospholipids increase twofold in the cortex (and threefold in the white matter) between the 10th week of gestation up until 2 years of age [35]. In this early work, we found a relative continuous decline in choline phosphoglycerides from 50% of total phospholipids in the cerebral cortex of the fetus to 45% in infants at term and 38% in children at 2 years of age. In addition, sphingomyelin showed an ongoing increase, from 3% of total phospholipids in the cerebral cortex of the fetus to 5% in infants at term and 10% amongst children at 2 years of age [35].

Choline deprivation during early development can lead to compromised cognitive function as the perinatal period is a critical time for cholinergic organization of brain function [36]. Early evidence from rodent models has played a central role in deciphering mechanisms and choline’s involvement in key processes affecting brain structure and function [22,37,38,39,40,41]. A review of 34 rodent studies [42] found that choline supplementation during gestation and the perinatal period: (1) enhanced cognitive performance—particularly on more challenging tasks; (2) increased (choline supplementation) or decreased (choline deficiency) the electrophysiological responsiveness and size of neurons in offspring; and (3) supplementation offered some protection against adverse effects of several neurotoxic agents (including alcohol) in offspring.

It is now known that choline is required by neural progenitor cells (starter cells) of the fetal hippocampus for membrane synthesis and methylation [8]. These, in turn, differentiate, migrate, proliferate, and undergo apoptosis at specific times during fetal development [8]. Choline modifies methylation of brain DNA and histones, which alter gene expression and encode proteins that play a role in memory and learning [43]. These underpinning mechanisms explain why, in rodent models, maternal dietary intakes of choline influences neurogenesis in the fetal hippocampus, and lead to life-long changes in memory function [8,44]. Such findings indicate that subsequent changes in memory function appear to be due to alterations in the memory center (hippocampus) of the fetal brain [45].

### 1.3. Accruement

During pregnancy, choline requirements rise due to increased maternal demands and rapid division of fetal cells. The availability of choline to cells will depend on food intake, its absorption, and the efficacy of cellular transport systems [46]. As de novo synthesis by the human body may become insufficient, choline is an essential dietary component [21]. So far, scientific observations have shown that plasma or serum choline levels are around 6 to 7-fold higher in the fetus and neonate than they are in adults showing demands for this nutrient [47,48]. The development of the central nervous system is particularly sensitive to choline availability with evidence of effects on neural tube closure and cognition [11].

Research using stable isotopes examining the effects of pregnancy on choline partitioning demonstrates enhanced use of choline for phosphatidylcholine production via both cytidine diphosphate-choline and PEMT pathways. This emphasizes that there are substantial demands for choline in the late stage (i.e., the third trimester) of pregnancy [28]. This preferential shuttling of PEMT-derived PtdCho during the third trimester may be attributed to its enrichment in docosahexaenoic acid (22:6n3), an essential omega-3 fatty acid that accumulates in the neonatal brain during the third trimester of pregnancy [49].

From a physiological stance, the last trimester of pregnancy is when the establishment of connections between brain regions within functional networks occurs [32]. Research investigating fetal brain maturation using proton magnetic resonance spectroscopy (^1^H-MRS) revealed that absolute metabolic concentrations of total choline (glycerol3-phosphocholine and phosphocholine) significantly increased between weeks 18 and 40 of pregnancy (*p* = 0.01) [50]. The human placenta also contains around 50 times more choline than the maternal blood (i.e., 1000 vs. 20 mol/L, respectively). It obtains choline predominantly via a saturable sodium independent carrier-mediated transport system that is both specific to and has a high affinity for choline [11,51,52].

Amongst preterm infants, plasma choline declines to 50% of cord plasma concentrations, indicating undernourishment and potentially contributing to impaired cognitive outcomes [53]. Human neonates are born with blood levels that are three times higher than maternal blood concentrations, and large amounts of choline are present in human milk [11]. Choline has been identified as one key nutrient affecting major brain processes, including neurogenesis, neuronal differentiation, myelination, and synaptogenesis, all of which proceed at a rapid pace between 22 and 42 weeks post-conception [54]. During infancy, myelination and synaptogenesis occur rapidly across the first 2–3 years of life [55] with the maturation (myelination) of these connections enabling the efficient transfer of information [32].

### 1.4. Choline Requirements and Dietary Intakes

Several leading organizations, including the American Medical Association and the Committee on Nutrition [56,57] and the American Academy of Pediatrics [58] now recognize that choline is a crucial nutrient during pregnancy and early childhood. Present dietary guidance by the European Food Safety Authority (EFSA) [21] advises an adequate intake (AI) of 400 mg/day for adults based on observed choline intakes in healthy populations. The AI is a recommended average daily intake based on observed estimations of nutrient intakes in healthy groups people. Ideally, a revision of reference intakes to include an estimated average requirement would significantly advance the field and better facilitate the understanding of dietary inadequacies [9,21]. Nevertheless, during pregnancy, the AI rises to 480 mg/day to account for increased gestational body weight [21]. For exclusive lactation, an intake of 520 mg/day have been set by EFSA, which provides an additional 120 mg/day on top of adult recommendations, that is, the estimated average amount of choline secreted in human milk daily [21]. Common genetic variants (such as those found in the PEMT and methylenetetrahydrofolate dehydrogenase (MTHFD1) genes) can also interact with choline metabolism and can impact on choline requirements [26].

Studies conducted in Europe [59,60] and the Americas [61,62] show that there is a tendency for average daily choline intakes to fall below the AI. For example, Vennemann et al. (2015) [59] using data from European surveys found average intakes to be 291–374 mg/day amongst females aged 18 to ≤ 65 years. Similarly, in the United States using NHANES (National Health and Nutrition Examination Survey) data, choline intakes for women of reproductive age were just 250 mg/day for females aged 19 to 30 years and 278 mg/day for those aged 31–50 years [21,61]. Among Latvian pregnant women, average choline intakes were 336 mg/day for adolescents and 356 mg/day for pregnant adults—both below European guidelines [60]. A prospective cohort among pregnant and lactating women in Alberta, Canada observed mean total choline intakes of 347 mg/day with just 23% and 10% of pregnant and lactating women meeting recommendations, respectively [62]. Other work concludes that around 90% of Americans, which includes most of the pregnant and lactating women, are substantially below the AI for choline [9,61]. This observation would suggest that large proportions of the populations have suboptimal choline intakes and/or status and leaves an open question of whether inadequate choline intake during pregnancy and lactation may influence neurodevelopment of the offspring.

### 1.5. Food Sources

With regard to food sources, choline can be present as both water-soluble (free choline, phosphocholine, and glycerophosphocholine) and lipid-soluble forms (phosphatidylcholine and sphingomyelin), though most surveys present this as ‘total choline’ [63]. Choline is predominantly present in animal foods containing fat and cholesterol; however, intakes of such foods are seemingly diminishing in response to health and environmental trends [8,64]. An analysis of 145 common foods using liquid chromatography-mass spectrometry shows that beef liver (418 mg/100 g), chicken liver (290 mg/100 g), eggs (251 mg/100 g), cooked salmon (90 mg/100 g), wheat germ (152 mg/100 g), bacon (125 mg/100 g), dried soybeans (116 mg/100 g), and pork (103 mg/100 g) have some of the highest choline profiles [12,65]. Amongst vegans, immature lima beans have been suggested as a favorable choline source [66]. Pulses provide some choline, although cooking methods appear to reduce the percentage of free choline and increase the contribution of phosphatidylcholine [67]. Subsequently, ongoing analysis is needed to compile accurate food composition data.

Looking at dietary contributions; eggs, fish, grains, meat, milk, and their relative derived products are some of the main choline providers [59]. In the Canadian Alberta study [62], pregnant women eating at least one egg during a 24-h period were eight times more likely to meet choline intake recommendations than non-consumers. Similarly, pregnant mothers obtaining ≥ 500 mL of milk over a 24-h period were 2.8 times more likely to meet choline dietary guidelines than those having less than 250 mL of milk daily [62]. Though many foods provide choline, it has been reported that humans have at least a twofold variation in dietary intakes [68]. Subsequently, trends away from animal-derived foods and towards plant-based and vegans diets could potentially impact choline intakes and status and have unintended consequences [3,64].

### 1.6. Choline Knowledge Gaps

There has been an emergence of interest in terms of how choline influences neurodevelopment and brain function. A large body of evidence has evolved over the last few decades from both animal studies and clinical trials [31,42,43,69]. From a health stance, in 2011, EFSA approved several claims for choline, including that choline consumption can: *“contribute to normal lipid metabolism”*, *“maintain normal liver functioning”*, and *“contribute to normal homocysteine levels”* [70].

Now approaching a decade, evidence for choline in relation to other aspects of health, including neurological development and brain health, has advanced. Consequently, the objective of this paper was to conduct a systematic review providing an update of evidence focusing on choline’s role in neurological development and brain function during the first 1000 days of life.

## 2. Materials and Methods

### 2.1. Approach

The present systematic review followed the Preferred Reporting Items for Systematic Reviews and Meta-Analyses (PRISMA) statement [71]. Abstracts and papers were selected for relevance. The computer software EndNotex9 was used during the selection process.

The National Institute of Health’s National Library of Medicine PubMed database was used to identify relevant publications studying associations between choline and brain function/neurodevelopment during the first 1000 days. In Phase 1, the search was restricted to animal studies published between January 1st 2010 and June 1st 2020, as earlier work has been reviewed elsewhere [30,42]. In Phase 2, a broader second search was undertaken to identify human trials published between January 1st 2000 and June 1st 2020. Reference lists of relevant publications were also searched for further studies.

The following terms formed the basis of the publication search: “choline AND brain”, OR “choline AND cognition”, OR “choline AND fetal”, “choline AND neuro*”, OR “choline AND preg*” OR “choline AND lact*”. The Boolean term AND was used to focus the search, while the wild card asterisk (*) was used to identify publications using the different terminologies.

### 2.2. Defining 1000 Days

The first 1000 days of life have increasingly become known as a ‘golden opportunity’ to shape a child’s cognitive, emotional, and behavioral development. This is because a great deal of the brains structure and capacity is formed by the age of 3 years [2]. Typically, the first 1000 days are defined as the maternal prenatal nutrition period and the child’s nutrition during the first 2 years of life [1,72].

While calories are needed for growth in early life, they are not required for ‘normal’ brain development. However, certain nutrients are essential for normal brain development, with choline being one of these [1]. Given this, the present review focuses on choline supplementation, intake, and status studies that were conducted in the prenatal period through to a child’s second birthday. With respect to animal studies, selected publications were restricted to a similar period when animal models were used.

### 2.3. Search Strategy

A PICO model formed the basis of the search strategy [73]. The population (P) was defined as the first 1000 days of life, which included pregnancy, lactation, neonates, and early life up to two years of age. The intervention (I) that was considered was choline supplementation, intake or status. The comparison (C) was a defined control or placebo group, and the outcomes of interest (O) were markers of brain function and neurological development.

Filters were applied. In Phase 1, the inclusion criteria were defined as: (1) articles published in English, (2) animal studies, (3) published within the last 10 years, and (4) papers studying the effects of choline supplementation, intake or status.

Phase 2 studies were required to satisfy the following inclusion criteria: (1) published in English, (2) human studies (clinical trials, randomized controlled trials, and observational studies), (3) published within the last 20 years, (4) studies focusing on the first 1000 days, and (5) papers investigating the effects of choline supplementation, intake or status.

Studies were included where choline was given alone or in combination with other micronutrients, but the impact of choline itself needed to be measurable and reported as an outcome. Studies were included when choline supplementation, intake or status were measured during a period aligning with the ‘1000 days’ definition. Papers were also considered if periods of follow-up were outside this period, e.g., a choline intervention in pregnancy but infant cognition assessed at 3 years. Publications that administered choline using daily injections in human studies were excluded.

### 2.4. Study Quality & Data Extraction

The Systematic Review Center for Laboratory animal Experimentation (SYRCLE) approach was used to determine the quality of animals studies [74]. This comprised six phases, that is, (1) developing the research question (does choline’s influence neurological development and brain function during the first 1000 days of life?), (2) searching for evidence, (3) selecting studies, (4) extracting study characteristics (Table 1), (5) assessing quality—the Hooijmans et al. (2014) [75] bias tool was applied, and (6) providing an overview of findings [74].

The Jadad scale and criteria were used to develop quality scores for each human RCT or clinical study [76]. Quality scores were graded between 1 and 5, with higher scores being indicative of higher quality. Studies with a Jadad score of <3 were regarded as being poor quality and not meeting international standards.

Papers were screened, and data was extracted using the predefined inclusion criteria. Once identified, all suitable publications had relevant data extracted. For the animal studies, the stage of the 1000 days, study methodology, study outcomes, and main findings were recorded. For the human studies, the author, year, the stage of 1000 days, study methodology (including dose and composition of choline supplements), study outcomes, and any other main findings were recorded.

## 3. Results

A total of 813 studies were identified—794 through database searches and 19 by additional manual searches for publications identified within the references of the found studies. After reviewing the full text, a total of 54 studies were selected—16 human studies and 38 animal studies (Figure 1). A narrative synthesis of the findings from the included studies was then performed. The synthesis was structured around the type of study and outcome.

### 3.1. Animal Studies

We identified 38 studies using animal models (mostly in rodents) to investigate the role of maternal choline supply in the brain and neurodevelopment of the offspring. In the last 10 years, three recent publications [77,78,79] used piglet models that represent a sensitive model for choline deficiency. Table 1 shows the studies organized according to the outcomes.

Most animal studies were relatively well designed. With regard to forms of bias, performance bias was most common amongst animal studies. For example, not all studies adequately reported how animals were housed, randomly allocated to groups, or blinded to the intervention [75]. Future studies also need to describe in more detail whether animals were selected at random to prevent detection bias as this was not always clear. Attrition tended to be well reported within studies, so attrition bias was less common.

#### 3.1.1. Brain Development

We identified eleven studies that had investigated the effects of choline supply on the development and structural organization of the brain (Table 1). Research that fed mouse dams either low-choline or control diets found that low-choline diets adversely modulated the development of the cerebral cortex [80]. In particular, the upper layer of cortical neurons decreased and two types of cortical cells—intermediate progenitor and radial glial cells—were reduced in fetal brains [80]. Elsewhere, choline supplementation by pregnant rats mitigated spatial learning deficits brought about by protein malnutrition by increasing synaptic cleft width and reducing the curvature of the synaptic interface in the hippocampus [81].

Other research using neuroimaging techniques on young pigs showed that prenatal choline deficiency reduced the gray matter in the left and right cortex, and white matter in the internal capsule and putamen compared with pigs that were ‘choline-sufficient’ prenatally [78]. Getty et al. (2015) [79] found that piglets fed a choline-deficient diet had smaller brains and reduced plasma concentrations of choline and choline-containing phospholipid levels than those who had received sufficient choline.

Multiple animal studies [82,83,84,85,86,87] have observed positive effects of choline on neurogenesis, many of which have studied effects on neuron gene expression. Effects of choline on angiogenesis have also been observed. A murine model [88] showed that maternal choline deficiency reduced the proliferation of endothelial cells in the hippocampus by 32% and reduced the number of blood vessels by around 25%.

#### 3.1.2. Cognition and Memory

Eight animal studies [89,90,91,92,93,94,95,96] considered the effects of choline sufficiency, or depletion in relation to aspects of behavior, learning, cognition, and memory performance. Moreno and de Brugada (2019) [96] using rodent models observed that those supplemented with choline prenatally spent less time exploring familiar objects, indicating that supplementation accelerated long-term memory development.

Jadavji et al. (2015) [90] identified short-term memory deficits, possibly from hippocampal dysfunction through enhanced apoptosis, in the offspring of dams subjected to dietary shortfalls of methyl donors, which included choline. Other work [92,93] demonstrates that prenatal choline-supplemented diets improve object recognition memory. Corriveau and Glenn (2012) [94] found that developmental choline levels helped to keep the memory intact in rats that were prenatally stressed using a model of schizophrenia.

Two studies focusing on iron deficiency have shown that deficits in recognition memory could be mitigated by prenatal choline supplementation, possibly by preserving hippocampal brain-derived neurotrophic factor and myelin basic protein expression [92]. Other research [97] using adult rats whose mothers were iron deficient during pregnancy and lactation found that choline administration during this period could act as a potential adjunctive therapy, attenuating the pathogenesis of neurological and psychological disorders.

#### 3.1.3. Protection from Environmental Exposures

Six animal studies using sheep, piglet, and rodent models have shown that choline supplementation, particularly over the prenatal period, could help mitigate some alcohol-induced neurodevelopmental effects [98,99,100,101,102,103].

In one study [99], rodents exposed to ethanol and fed a low choline diet (40% of recommended choline intake levels) had delayed eye openings, poor hind limb coordination, and were overactive compared to those fed 70% or 100% recommended choline levels, indicating that suboptimal intakes can exacerbate the teratogenic effects of ethanol. Bekdash et al. (2013) [103] found that gestational choline delivered to pregnant rat dams prevented adverse effects of prenatal ethanol exposure on neurons. Another rat model [104] showed that alcohol exposure in utero could result in long-lasting changes to the hippocampal cholinergic system, which was attenuated by choline supplementation. Similarly, Thomas et al. (2010) [101] using a rodent model found that choline intubation attenuated the effects of prenatal alcohol exposure, especially concerning tasks that required behavioral flexibility.

In one sheep model, maternal choline supplementation at levels resembling doses used in human studies (10 mg/kg/day choline) failed to prevent brain volume reductions induced by first-trimester binge alcohol exposure [105]. In contrast, another sheep model investigating fetal cranio-facial abnormalities found that maternal choline supplementation (10 mg/kg in the daily food ration) mitigated the effects equivalent to a first-trimester alcohol binge and additionally improved fetal femur and humerus bone lengths [98].

#### 3.1.4. Choline Supply in Animal Models of Neurodegeneration

We identified nine studies that were published since 2010, which investigated the effects of choline supply on aspects of memory, learning, and cognitive function using mouse models of Down Syndrome and Alzheimer’s disease [83,84,85,86,95,106,107,108,109].

Alldred et al. (2019) [82] found that maternal choline supplementation in a mouse model of Alzheimer’s disease and Down syndrome favorably altered the expression of genes involved in GABAergic neurotransmission and neurotrophins. This highlights the importance of adequate choline intake, particularly when the fetus has a neurodevelopmental disorder such as trisomy. Earlier work [106] also found that maternal choline supplementation normalized the expression of several genes in a murine model of Alzheimer’s disease and Down syndrome, including those involved in calcium-signaling, synaptic plasticity, and Alzheimer’s disease-associated neurodegeneration.

Kelley et al. (2019; 2014) [83,85] using murine models found that maternal choline supplementation attenuates some of the genotype-dependent alterations in the basal forebrain cholinergic neuron (BFCN) system. Other work [108] by the same team found that increased innervation produced by maternal choline supplementation improved hippocampal function. Choline supplementation (5.0 g/kg choline chloride) was associated with increases in the density, size, and the number of medial septum BFCNs compared to a choline-sufficient diet (1.1 g/kg choline chloride). This indicated that this could be a safe and viable treatment option for mothers carrying a Downs Syndrome fetus which is typically associated with BFCN degeneration [84].

Focusing on the longer-term effects of maternal choline supplementation during the perinatal period, Yan et al. (2014) [109] using a trisomic mouse model of Down syndrome and Alzheimer’s disease found that adult offspring of choline supplemented versus unsupplemented dams had 60% greater PEMT activity. This appeared to have lasting effects, including the enhanced provision of choline and PEMT-PC (enriched in long-chain unsaturated fatty acids) to the brain.

Another murine model of Down Syndrome and Alzheimer’s disease [86] found that Ts65Dn offspring of choline-supplemented dams performed significantly better on spatial cognition tests than non-supplemented dams, indicating translational benefits. Similarly, Moon et al. (2010) [95] found that perinatal choline supplementation reduced cognitive dysfunction in trisomic mice.

Velazquez et al. (2019) [107] studied the effects of maternal choline supplementation across two generations of mice. They found that maternal choline supplementation improved cognitive deficits and reduced amyloid beta accumulation load, microglia activation, and Alzheimer’s disease pathology across the two generations, which were possibly linked to reduced brain homocysteine levels.

**Table 1 nutrients-12-01731-t001:** Animal studies investigating the inter-relationships between choline, brain function and neurological development.

Study (Author, Year, Country)	Animal Study	Stage of 1000 Days	Study Methodology	Study Outcomes	Main Findings
Alldred et al. (2019) [82] USA	Mouse model of DS and AD	Pregnancy	Perinatal choline supplementation.	Neuron gene expression	Maternal choline supplementation increased offspring gene expression.
Chin et al. (2019) [110] Singapore	Knockout mice	Postnatal	Fed 13 mg/day (1.7 × required daily intake) of choline	Motor co-ordination, behavioral deficits, anxiety	Choline modulated neuronal plasticity, leading to behavioral changes and showing potential to treat RTT.
Kelley et al. (2019) [83] USA	Mouse model of DS and AD	Pregnancy	Dams on an MCS diet or a normal choline diet from mating until weaning,	Neuron gene expression	Significant downregulation was seen in select transcripts that were normalized with MCS.
Moreno & Brugada et al. (2019) [96] Spain	Rat model	Pregnancy	Fed with 1.1 g choline/Kg food or 5 g choline/Kg food between embryonic days (E) 12 and E18.	Long-term memory	Prenatal supplementation with choline accelerates the development of long-term memory in rats.
Sawant et al. (2019) [98] USA	Sheep model	Pregnancy (1st Trimester)	Randomized to six difference ethanol/choline groups.	Alcohol-induced fetal cranio-facial abnormalities	Maternal choline supplementation mitigated most alcohol-induced effects.
Velazquez et al. (2019) [107] USA	Mouse model of AD	Transgenerational effects of MCS	Exposed mice to MCS and bred for two generations.	Alzheimer’s disease, brain homocysteine	Providing MCS reduced AD pathology across two generations.
Alldred et al. (2018) [106] USA	Mouse model of DS and AD	Pregnancy	Provided with MCS.	Neuron gene expression	MCS reprogrammed transcripts involved in neuronal signaling.
Kennedy et al. (2018) [89] USA	Male rat pups with ID	Pregnancy, nursing and early life	Choline (5 ppm) was given to half the nursing dams and weanlings	Cognitive performance, novel object recognition	Recognition memory deficits induced by early-life iron deficiency was prevented by postnatal choline supplementation.
Mudd et al. (2018) [78] USA	Pigs (Yorkshire sows)	Pregnancy and nursing	Choline-sufficient or choline-deficient diet or milk supply	Brain white and grey matter	Prenatal choline deficiency greatly alters grey and white matter development in pigs. No postnatal effects were observed.
Balaraman et al. (2017) [100] USA	Sprague-Dawley rats	Postnatal (+ethanol exposure)	Treated with choline chloride (100 mg/kg/day) or saline	Hippocampal microRNA alterations	Choline supplementation can normalize disturbances in miRNA expression following developmental alcohol exposure.
Idrus et al. (2017) [99] USA	Rat model	Pregnant	Received diets containing 40, 70, or 100% recommended choline levels	Motor development, co-ordination	Subjects exposed to ethanol and fed the low 40% choline diet had delayed eye openings, poor hind limb coordination, and were overactive compared to all other groups.
Mellott et al. (2017) [111] USA	Rat model	Pregnancy and nursing	Fed a diet containing 1.1 g/kg of choline or a choline-supplemented (5 g/kg) diet.	Amyloidosis, hippocampal choline acetyltransferase expression	The choline group had reduced levels of solubilized amyloid peptides and plaques; preserved levels of choline acetyltransferase protein and absence of astrogliosis indicating a role in AD prevention.
Birch et al. (2016) [105] USA	Sheep model	First trimester (alcohol binge)	Randomly assigned to: HBA HBC (2.5 g/kg ethanol and 10 mg/kg/day choline), saline control, saline control plus choline (10 mg/kg/day choline), and normal control.	Brain volume	Maternal choline supplementation comparable to doses in human studies fails to prevent brain volume reductions typically induced by first-trimester binge alcohol exposure.
Kelley et al. (2016) [108] USA	Mouse model of DS	Pregnancy	Provided with MCS.	Hippocampal function	Maternal choline supplementation increased innervation and improved hippocampal function.
Mudd et al. (2016) [77] USA	Piglets from Sows	Perinatal period	Choline-sufficient or choline-deficient diet or milk supply	Brain development	Prenatal choline deficiency had profound effects by delaying neurodevelopment as evidenced by reduced concentrations of glycerophosphocholine-phosphocholine, brain volumes and region-specific volumes.
Tran et al. (2016) [97] USA	Rat model	Pregnant and nursing dams fed an ID diet	Choline (5 g/kg) was given to half the pregnant dams in each group	Hippocampal function	Choline supplementation reduced the effects of ID, including those on gene networks associated with autism and schizophrenia.
Wang et al. (2016) [80] USA	Mouse model	Pregnancy	Fed either control or low-choline diets	Cortical development	Low choline supply reduced the number of 2 types of cortical neural progenitor cells, radial glial cells and intermediate progenitor cells in fetal brains (*p* < 0.01).
Zhu et al. (2016) [81] China	Rat model	Pregnant	Fed a normal or low-protein diet containing sufficient choline (1.1 g/kg choline chloride) or supplemented choline (5.0 g/kg choline chloride) until delivery	Spatial learning deficits	Prenatal choline supplementation reversed the increased width of the synaptic cleft (*p* < 0.05) and decreased the curvature of the synaptic interface (*p* < 0.05) induced by a low-protein diet.
Bearer et al. (2015) [102] USA	Mouse model	Pregnant	Maintained a choline-deficient diet and the 1 of 8 treatments	Balance and co-ordination	Choline alleviated ethanol-induced effects on balance and co-ordination.
Getty et al. (2015) [79] USA	Sows	Pregnancy and nursing	Fed a choline deficient or sufficient diet and milk supply	Brain development	The brains of piglets exposed to prenatal choline deficiency were significantly smaller than those of choline sufficient piglets.
Jadavji et al. (2015) [90] USA	Mouse model	Pregnancy and nursing	Effect of maternal choline deficiency	Short-term memory, apoptosis	There were short-term memory deficits in the offspring of dams with dietary deficiencies of critical methyl donors, i.e., choline.
Langley et al. (2015) [91] USA	Mouse model of autism	Pregnancy and nursing	Fed a control or choline-supplemented diet from mating	Social interaction and anxiety	High choline intake during early development reduced deficits in social behavior and anxiety in an autistic mouse model.
Ash et al. (2014) [84] USA	Mouse model of DS	Pregnancy and nursing	Assigned to a choline sufficient (1.1 g/kg choline chloride) or choline supplemented (5.0 g/kg choline chloride) diet.	Basal forebrain cholinergic neuron number and size	Maternal choline supplementation significantly improved spatial mapping and increased number, density, and size of MS BFCNs in DS offspring.
Kelley et al. (2014) USA	Mouse model of DS	Pregnancy	Studied effects of MCS	Basal forebrain cholinergic system	MCS partially normalized the BCFN system.
Kennedy et al. (2014) [92] USA	Male rat pups with ID	Pregnancy	Provided with choline supplementation (5 g/kg choline chloride, E11-18) or control	Neurobehavioral effects	Prenatal choline supplementation in formerly ID rats restored novel object recognition and increased hippocampal gene expression.
Yan et al. (2014) [109] USA	Mouse model of DS and AD	Pregnancy	Previously mothers had been choline-supplemented	PEMT pathway	Maternal choline supplementation upregulates PEMT pathway and d9 choline metabolites in the brain.
Schulz et al. (2014) [112] USA	Rat model	Pregnancy and nursing	Control and stressed dams were fed choline-supplemented or control chow	Anxiety	Perinatal choline supplementation mitigated prenatal stress-induced social behavioral deficits in males.
Bekdash et al. (2013) [103] USA	Rat model	Pregnancy	Fed an alcohol-containing liquid diet or control diet with or without choline	Neuronal function	Gestational choline supplementation prevents the adverse effects of ethanol on neurons.
Moreno et al. (2013) [93] Spain	Rat model	Pregnancy	Fed choline-deficient (0 g/kg choline chloride), standard (1.1 g/kg choline chloride), or choline-supplemented (5 g/kg choline chloride) diets	Memory	The supplemented group exhibited improved memory compared with both the standard and the deficient group.
Velazquez et al. (2013) [86] USA	Mouse model of DS and AD	Pregnancy	Fed additional choline (4.5x more than normal)	Hippocampal neurogenesis	MCS partially normalized adult hippocampal neurogenesis.
Corriveau & Glenn (2012) [94] USA	Rat model	Postnatal	Rats fed a choline-supplemented, -deficient, or standard diet	Cognitive functioning	Choline deficiency impaired memory in rats that were stressed prenatally.
Monk et al. (2012) [104] USA	Rat model	Perinatal	Injected with choline chloride (100 mg/kg/day) or saline vehicle	Hippocampal cholinergic development	Perinatal choline supplementation can attenuate alcohol-related behavioral changes by influencing cholinergic systems.
Otero et al. (2012) [113] USA	Rat model	Neonatal period	Choline or saline administered subcutaneously	DNA methylation in the hippocampus and prefrontal cortex	Alcohol exposure induced hypermethylation in these regions with was significantly reduced after choline supplementation.
Wong-Goodrich et al. (2011) [114] USA	Rat model	Prenatal	Received either a control or choline supplemented diet	Long-term cognitive and neuropathological effects	Prenatal choline supplementation promoted long-term hippocampal recovery from seizures in adulthood.
Mehedint et al. (2010a) [88] USA	Mouse model	Pregnancy	C57BL/6 mice were fed either a choline-deficient, control or choline-supplemented diet.	Angiogenesis	Maternal dietary choline intake altered angiogenesis in the developing fetal hippocampus.
Mehedint et al. (2010b) [87] USA	Mouse model	Pregnancy	C57BL/6 mice were fed either a choline-deficient, control or choline-supplemented diet.	Methylation and epigenetic marking	Choline deficiency altered histone methylation in neural progenitor cells which appears to underlie the observed changes in neurogenesis.
Moon et al. (2010) [95] USA	Mouse model of DS and AD	Perinatal	Choline-supplemented Ts65Dn dams	Cognitive functioning	Perinatal choline supplementation may lessen cognitive dysfunction in DS and reduce cognitive decline in related disorders such as AD.
Thomas et al. (2010) [101] USA	Rat model	Pregnant	Intubated with either 250 mg/kg/day choline chloride or vehicle	Working memory, behavior	Choline supplementation during prenatal alcohol exposure may reduce the severity of fetal alcohol effects, particularly on alterations in tasks that require behavioral flexibility.

Key: AD, Alzheimer’s disease; BFCNs, Basal forebrain cholinergic neurons; DNA, deoxyribonucleic acid; DS, Down Syndrome; HBA, heavy binge alcohol; HBC, heavy binge alcohol plus choline supplementation; ID, iron deficiency/deficient; MCS, maternal choline supplementation; MS, medial septum; PEMT, phosphatidylethanolamine N-methyltransferase; RNA, ribonucleic acid; RTT, Rett syndrome.

### 3.2. Human Studies

#### 3.2.1. Fetal and Infant Neurodevelopment

We identified 16 human studies that looked at choline in relation to aspects of neurological development and brain function (Table 2). Of these, 13 focused on choline and aspects of fetal or infant neurodevelopment without the presence of any insults [115,116,117,118,119,120,121,122,123,124,125,126,127] and three studies were conducted with alcohol-exposed pregnant women or in infants with prenatal alcohol or infection exposures [128,129,130].

Choline interventions started in pregnancy, typically the second or third trimester, although three studies Ross et al. (2016) [118], Boeke et al. (2013) [120], and Vilamor et al. (2012) [123] commenced earlier. Nine of the identified studies were RCTs [115,116,117,118,119,121,122,129,130] and seven were prospective cohort, cross-sectional, or case-control studies [120,123,124,125,126,127,128]. Eight studies were regarded as being of moderate to high quality using the Jadad criteria (scoring > 3) [115,116,117,118,119,121,122,129]. Four studies were rigorously designed and scored five (high quality) [116,117,122,129] while one was poor quality [130], and another described the methodology in a different paper, making its design unclear [115] (Table 3).

Caudill and colleagues [117] conducted a randomized, double-blind controlled feeding study with 26 pregnant women in their third trimester (27 weeks gestation) randomized to consume 480 or 930 mg choline/day (380 mg choline/day from diet and either 100 or 550 mg/day from supplemental choline) until delivery. Study findings revealed faster information processing speed amongst infants whose mothers consumed twice 930 versus 480 mg choline/day during the last trimester of pregnancy. Notably, in the 480 mg choline/day group, a longer duration of fetal exposure to this level of maternal choline intake was also related to faster infant processing speed [117]. When these infants were followed up at 7-years of age (106), children from the 930 mg choline/day group reached higher levels on computerized color-location memory tasks than the lower-dose choline group, indicating superior memory span [115].

These findings are broadly consistent with that from other trials. For example, Ross et al. (2013) [121] randomized 100 healthy pregnant women to supplement with twice the normal choline levels (≈900 mg choline/day) from the second trimester. Results showed that auditory sensory gating improved amongst 5-week-old infants born to choline-supplemented mothers [121]. In a later follow-up from this study, 40-month-old infants born had fewer attention problems and were less socially withdrawn in the phosphatidylcholine (900 mg/day) group as compared to the control group [118]. These effects could have been mediated by phosphatidylcholine increasing activation of the α7-nicotinic acetylcholine receptors with the authors concluding that this could alter the development of behavior problems in early life that may presage poor mental health [118].

Likewise, observational evidence demonstrates associations between maternal choline intake and infant cognition, intelligence, and memory [120,124,125,126]. Findings from the US Project Viva cohort found that higher choline intakes in pregnancy (mean second-trimester intake; 328 mg/day) associated with improved child visual memory at age 7-years [120]. A prospective Canadian cohort [124] showed that higher levels of plasma free choline in the first half of pregnancy (16 weeks gestation) associated positively with infant neurodevelopment and cognitive test scores (*p* = 0.009). Two other prospective cohorts reported similar inter-relationships [125,126]. A large Californian cohort [125] comprised of 180,000 women observed that higher mid-pregnancy serum total choline levels were associated with a reduced risk of neural tube defects (NTDs), indicating a protective role of choline. Similarly, in an earlier study [127], choline intake in the periconceptional period (defined as the 6 months from 3 months before to 3 months after conception) associated with reduced NTD risk.

Two trials [116,122] and observational studies [123,126] found no inter-relationships between maternal choline supply and offspring intelligence, cognition nor neurocognition [116,122,123,126]. Cheatham et al. (2012) [122] randomly allocated 140 pregnant women to receive 750 mg/day PtdCho or a control from 18 weeks gestation until 90 days postpartum. Infants’ global and language development, short and long-term visuospatial memory did not significantly improve, with the authors concluding that the short follow-up period and lower supplement dose could have contributed to the lack of findings [122]. Later on, in the postnatal period Andrews et al. (2018) [116] randomized infants with neurological impairment risk factors to receive a treatment (providing 10.5 mg/day choline) or control supplement for 2 years with no significant neurodevelopmental benefits observed.

Concerning observational evidence, data extracted from 404 maternal-child pairs from the US infant growth project [126] showed that neither pregnancy nor new-born choline levels within physiologic ranges were associated with childhood intelligence. Villamor et al. (2012) [123] found no significant association between maternal choline intakes in the first or second trimesters and childhood cognition at age 3-years. However, it was noted in both studies that choline supplies were based on regular dietary intakes which may not have been high enough to induce measurable effects [123,126].

#### 3.2.2. Protection from Infections and Alcohol Exposure

Four human studies focused on whether maternal choline supplementation plays a role in ameliorating certain neural insults [119,128,129,130]. One trial [128] recruited pregnant mothers with infection(s) at 16 weeks and observed that those with higher serum choline concentrations delivered new-borns with better inhibition of auditory cerebral response and improved development of self-regulation, i.e., were more able to regulate their emotions and behavior at 1-year of age [128].

Other studies [119,129,130] focused on the effects of choline interventions given to pregnant heavy alcohol/ethanol drinkers and the implications for infant brain development, cognition, and ability to process information. One randomized trial [129] conducted in Cape Town South Africa allocated heavy drinkers to an oral dose of 2 g choline/day from mid-pregnancy until delivery and reported mitigation of some of the adverse effects of prenatal alcohol exposure on infant cognitive function and growth [129]. The choline-treated infants had better catch-up growth in terms of weight and head circumference at 6.5 and 12 months, and improved visual recognition memory at 12-months of age [129].

A further two studies took place in Western Ukraine [119,130]. In one trial [130] moderate to heavy drinkers who took a multivitamin/mineral supplement (providing 750 mg/day choline) during pregnancy had infants who performed better on infant development tests at 6-months of age. Another publication by the same team of scientists also reported improved processing skills among the infants of the supplemented mothers [119]. In attempting to explain the findings, authors proposed that choline supplementation may positively influence brain development by preventing fetal alcohol-related depletion of dimethylglycine—a metabolic nutrient that can safeguard against overproduction of glycine, during critical periods of neurogenesis [119]. Glycine receptors are expressed prominently in the brain during the early phases of development and pathological consequences resulting from their dysfunction may offer some explanation to syndromes that undermine brain function [131].

## 4. Discussion

Choline has recently been coined a ‘neuroprotectant’ and ‘neurocognitive essential nutrient’ that is critical for normal growth and functioning of the early brain [9,21,132]. Despite the enhanced capacity to synthesize choline during critical life-stages such as pregnancy and lactation, fetal/infant demand is so high that this exceeds de novo synthesis by the human body in this time, thereby making it an essential component of the diet [21]. Presently, ≈90% of Americans have choline intakes falling below the basic AI, including most pregnant and breastfeeding mothers (AI of 450 and 550 mg/day, respectively) [3,61]. Contemporary shifts towards plant-based diets and veganism are also likely to exacerbate further the risk of choline inadequacies [3,64], meaning that women are at ‘high risk’ of entering conception with suboptimal choline intakes and/or status. This is particularly concerning given that choline supplies that are shown to be beneficial to infant memory and processing speed were up to 930 mg/day [115,117]—nearly double the AI which itself is generally underachieved.

The present publication conducted a systematic review of evidence from 38 animal and 16 human studies, which indicated a consistent body of evidence that choline plays a significant role in proper neurodevelopment and brain function over the first 1000 days of life. Evidence from animal models shows that higher choline intakes during pregnancy, and in early postnatal development, can protect the brain from the neurological damage associated with fetal alcohol syndrome and inherited conditions such as Down Syndrome. It also has lasting effects in adulthood, including improved cognitive function, prevention of age-related memory decline, and neurological changes linked to conditions such as Alzheimer’s disease [31,43,69]. Evidence from a growing body of human studies also shows that choline interventions in pregnancy can improve infant processing speed and visuospatial memory in childhood [115,117,119], indicating that it could be time to update health policies and nutrient guidelines to help better communicate such findings to public sectors. The majority of studies focused on pregnancy, particularly from mid-pregnancy, which signifies this to be a particularly sensitive period for the functional effects.

In conducting the present systematic review, several significant findings are worthy of contemplation. Firstly, there is a substantial body of evidence to signify that choline can help to support normal brain development. For example, several recent high-quality RCTs have reflected important findings in this field. Bahnfleth et al. (2019) [115] concluded that the offspring of mothers who ingested 930 (vs 480) mg choline daily (100 and 550 mg choline/day as supplementation) from 27 weeks gestation until delivery performed significantly better at 7-years of age when tested on color-location memory. The result implies a long-term beneficial effect role of prenatal choline supplementation on offspring cognition. Caudill et al. (2018) [117] also demonstrated that maternal consumption with around twice the recommended amount of choline—100 and 550 mg choline/day as supplementation on top of a dietary intake providing 380 mg choline/day (thus total choline was 480 or 930 mg/day) improved infant processing speed when taken from the third trimester of gestation. Ross et al. (2016) [118] showed that perinatal phosphatidylcholine supplementation (900 mg choline/day) from the second trimester of pregnancy until birth reduced attentional problems and social withdrawal in 40-month-old children in the intervention group versus control.

In addition, the research of Jacobson et al. (2018) [129] is one of the first to show that high dosages of supplemental choline (2 g daily), when administered early in pregnancy, attenuated the effects of heavy prenatal alcohol exposure on cognition, eyeblink conditioning, and postnatal growing in the offspring born to these mothers. Similarly, observational evidence conducted in 2013 [120] using US Project Viva data showed that higher gestational choline intakes during the second trimester (median intakes of 392 mg/day versus 260 mg/day) correlated positively with modestly better child visual memory at age 7 years. Results for the first-trimester intake followed the same trend but were weaker [120]. Together, these findings yield an updated perspective that adequate maternal choline intake during pregnancy has the potential to influence brain development, cognition, and other related outcomes positively.

Secondly, it should be considered that around 44% of pregnancies globally are presently unintended (≈30% in developed countries) [133]. Alcohol is also a well-established teratogen that can have both physical and behavioral effects on the fetus [134]. In the US, the prevalence of alcohol use and binge drinking amongst non-pregnant women is approximately 53.6% and 18.2%, respectively [135]. In pregnancy, higher frequencies and amounts of binge drinking have also been reported than in nonpregnant women [135]. From the evidence-base, it is increasingly apparent that choline can attenuate some of the neurological damage associated with fetal alcohol syndrome [31,98,99,101,102,103,104,113,119,129]. Women need to be better informed that there is no known safe level of alcohol consumption when they might be or are pregnant. In addition, all women who consume alcohol and could potentially conceive should consider improving their choline profiles from a protective perspective. Further work has identified important associations between maternal choline profiles and reduced risk of neural tube defects [125,127]. It is also known that SNPs in folate-mediated pathways could mean that individuals with such genotypes would benefit from choline intakes which exceed present recommendations [136]. For example, such individuals have been found to have a loss-of-function in certain folate-metabolizing enzymes, placing a strain on PC production [136]. This further reinforces the significant role of choline during the childbearing years and its ability to attenuate unfavorable and potentially harmful metabolic insults.

Thirdly, there is now a well-established body of evidence from animal studies showing that higher choline intakes can ameliorate neurological damage associated with inherited conditions such as Down Syndrome, as well as potentially protecting the brain from the neuropathological changes associated with Alzheimer’s Disease [43,82,83,84,85,86,95,106,107,108,109,137]. In particular, the scientific evidence shows that maternal choline supplementation protects basal forebrain cholinergic neurons and normalizes neurogenesis helping to improve cognition and attention in individuals with Down Syndrome, who can exhibit hallmarks of Alzheimer’s Disease as early as the third decade of life [137,138]. Thus, it appears that choline could lessen intellectual disability by improving neural and cognitive functioning. The next stage of research would be to develop human trials in this field and build on findings from animal models.

Higher choline intakes during the first 1000 days’ life stage could also have other, extended benefits. For example, there is evidence to suggest that maternal choline supplementation has protective effects against lipopolysaccharide-induced inflammatory responses [139]. Other studies [140,141,142] demonstrate potential roles relating to the modulation of placental nutrient transport and metabolism, placental epigenome and placental markers of apoptosis, inflammation, and vascularization. King et al. (2019) [143] recently observed that increasing choline supply over 12 weeks in pregnant women led to higher levels of holotranscobalamin (the bioactive form of vitamin B12), indicating that choline supplementation could beneficially modulate vitamin B12 status in pregnancy. Additional work [144,145] suggests that maternal choline supplementation helps to normalize fetal growth and adiposity in the offspring of obese mice, possibly by altering the expression of genes involved in de novo lipogenesis and subsequent lipid metabolism.

Growing evidence indicates that choline supply may favorably influence functional processes of the placenta, including angiogenesis, inflammation, and macronutrient transport [69]. A 12-week trial which randomized 26 healthy pregnant women to take 480 mg or 930 mg/day choline for 12-weeks from 26–29 weeks gestation showed that there was statistically significant (30%) down-regulation of the antiangiogenic factor and preeclampsia risk marker fms-like tyrosine kinase-1 in the placenta tissues. The result indicated that supplementing the maternal diet with extra choline could improve placental angiogenesis and allay some of the pathological antecedents of preeclampsia [146]. Other work building on these findings cultured immortalized HTR-8/SVneo trophoblasts in different choline concentrations and found that choline insufficiency altered the angiogenic profile, impaired in vitro angiogenesis, increased inflammation and oxidative stress, induced apoptosis, and produced greater levels of protein kinase C isoforms δ and ϵ. This indicated that choline shortfalls could contribute to placental dysfunction [147]. Further to this, rodent data also shows that prenatal choline supply could be an effective nutritional approach to attenuate placental insufficiency [140,148].

Finally, it is important to recognize research limitations and mention potential future research directions. Nine out of the 16 human studies were RCTs, and eight of these were average-high quality, indicating an ongoing need for high-quality trials. It is well recognized that RCTs are the ‘gold standard’ as cause and effect relationships are more challenging to determine from observational studies [149,150]. Alongside the need for ongoing RCTs, larger sample sizes, longer time frames, multiple and aligned dosages of choline would benefit future trials. It is also important to use clearly defined endpoints to assess neurological development and brain function so that study comparisons can be made effectively and future meta-analytical work undertaken. As mentioned, SNPs could also influence the choline requirements of pregnant women and should be considered when designing future studies and compiling public health guidance [26,27].

In instances where significant findings were not observed, this could be attributed to the insensitivity of tests or the possibility that effects were too small to detect in some studies [130]. It is also plausible that the effects of choline on markers of brain function, such as intelligence, were overshadowed by the more significant influence of social and economic factors [126]. Participant adherence to choline programs should also be monitored and carefully reported within each study. In some studies, higher doses of choline were not used, and it could be that these were required to generate measurable effects. Extended intervention periods, beyond the third trimester and prolonged periods of follow-up would also help to capture better any benefits related to maternal choline supplementation [117].

In the light of growing evidence, it seems tenable to suggest that most women of childbearing age who could fall pregnant would benefit from increased intakes of choline. It appears prudent that health policies and dietary guidelines are updated to reflect the growing weight of evidence in this field. To reiterate, the EFSA presently advises a choline AI of 400 mg/day for non-pregnant adults, 480 mg/day for pregnancy and 520 mg/day for exclusive lactation [21]. EFSA considers this higher AI during pregnancy based on choline accretion in the fetus and placenta during this critical time and to facilitate the transfer of long-chain polyunsaturated fatty acids to the fetus via PtdCho in lipoproteins to support brain development [21]. Lactating women may need more choline than non-lactating women as breast milk is rich in choline and increasing maternal choline intake enhances the concentration of choline in breast milk [21].

Most human studies use supplements providing up to 930 mg choline daily and suggest potential benefits for offspring memory and attention, with no adverse effects reported [115,117,118,121]. Ongoing dose-response studies are needed, but it could be questioned whether present guidelines are high enough to support optimal neurological development of the offspring. Therefore, this raises concerns and questions about whether choline dietary guidelines require revision. Thus, in the meantime, as concluded by Korsmo et al. (2019) [69], the consumption of 450–1000 mg choline/day appears to be a suitable intake level that would support fetal brain function and neurodevelopment, alongside other pregnancy outcomes. Finally, from here on, it is important to impart ongoing awareness to healthcare professionals, including midwives, obstetricians, and pediatricians about the importance of choline across the first 1000 days of life.

## 5. Conclusions

Collectively, the evidence shows that choline is an essential ‘neurocognitive nutrient’ that has a pivotal role in proper neurological and brain development. There is now an extensive body of evidence from animal studies, which is increasingly being reinforced by human trials, suggesting three overarching pillars of potential benefit for choline supplementation programs. The pillars are: (1) supporting normal brain development, (2) protecting against alcohol exposure and infections, and (3) dampening intellectual disability by improving neural/cognitive functioning and memory. Given shifting dietary trends away from choline-rich foods, now seems to be an appropriate time to reflect on present dietary guidance and raise awareness about choline’s important roles across the crucial first 1000 days of life, especially concerning neurological development and brain function.

## Figures and Tables

**Figure 1 nutrients-12-01731-f001:**
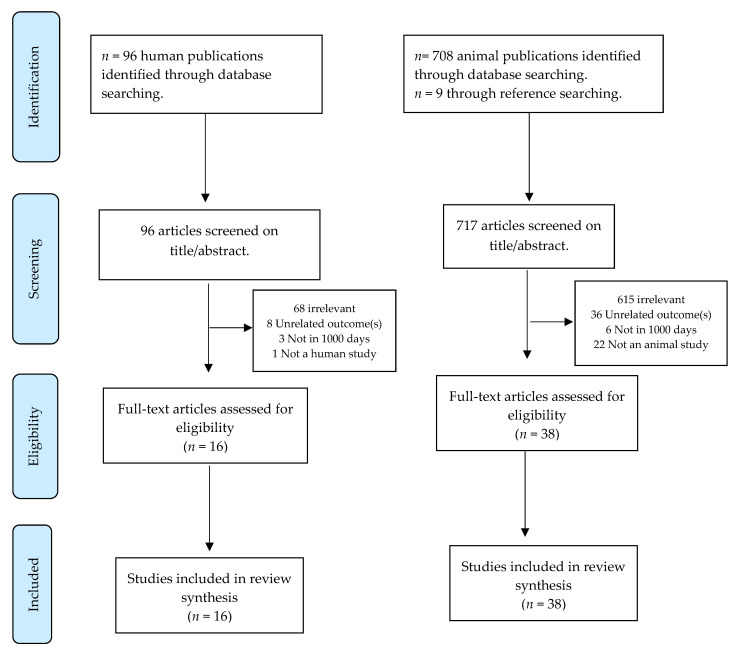
Preferred Reporting Items for Systematic Reviews and Meta-Analyses (PRISMA) algorithm used to identify studies.

**Table 2 nutrients-12-01731-t002:** Human studies investigating inter-relationships between choline, brain function and neurological development.

Study (Author, Year, Country)	Stage of 1000 Days	Study Design	Dose of Choline/Choline Measurements	Study Outcomes	Main Findings
Bahnfleth et al. (2019) [115] USA	3rd T (from 27 weeks gestation for 12 weeks) (*n* = 20)	DB randomized parallel-group controlled feeding intervention.	Randomized to consume 480 mg or 930 mg choline/d from gestational week 27 until delivery.	Computer-based color-location memory task	Children whose mothers consumed 930 vs.480 mg choline/day performed significantly better on a task of color-location memory at age 7 years, suggesting a long-term beneficial effect of prenatal choline.
Freedman et al. (2019) [128] USA	2nd T. Mothers with an infection from 16 weeks gestation and their infants (*n* = 66)	Prospective Cohort	Serum-free choline and baseline levels.	Infant brain development, cerebral inhibition, and auditory response	Development of cerebral inhibition, auditory cerebral response and behavioral regulation significantly improved in infants born to infected mothers with higher gestational choline concentrations, attenuating the effects of infections.
Andrew et al. (2018) [116] UK	Infants aged 1 to 18 months (*n* = 40)	DB RCT.	Randomized to a supplementation program (included 10.5 mg choline) or control.	Neurodevelopment	No statistically significant differences in neurodevelopmental outcome were identified between groups.
Caudill et al. (2018) [117] USA	Entering 3rd T. Infants assessed up to 13 months (*n* = 24).	DB randomized parallel-group controlled feeding intervention.	Choline supplement of either 100 or 550 mg/day. Diet provided 380 mg choline/day. Total choline received was 480 or 930 mg/day.	Infant processing speed, visuospatial memory	Mean reaction time was significantly faster for infants born to mothers in the 930 (vs. 480) mg choline/day group. Maternal consumption of approximately twice the recommended amount of choline during the last trimester improved infant information processing speed.
Jacobson et al. (2018) [129] South Africa	Mid-pregnancy—heavy drinkers (*n* = 69).	DB PC RCT.	2 g of choline daily (choline bitartrate) or placebo from enrolment until delivery.	Infant cognitive function, eyeblink conditioning	At 6.5 months infants in the choline arm had better eyeblink conditioning and at 12 months higher novelty preference scores, indicating better visual recognition memory.
Ross et al. (2016) [118] USA	1st T. Data for *n* = 49 infants.	PC RCT.	Received 900 mg choline/day. After birth, infants received 100 mg of phosphatidylcholine in an oral suspension once daily or placebo.	Childhood behavior, attention problems	At 40 months, parent ratings of children in the phosphatidylcholine group indicated fewer attention problems and less social withdrawal compared with the control group.
Coles et al. (2015) [130] USA	2nd T (from 19 weeks gestation). Moderate to heavy drinking (*n* = 301), and low/unexposed comparison women (*n* = 313).	Prospective cohort study (and RCT within)	Randomly assigned to receive: (1) daily MVM supplement, (2) “standard of care” or (3) MVM-supplement providing 750 mg choline/day.	Infant development (Bayley Scales)	Developmental improvement in infants associated with choline seen not observed in this study.
Kable et al. (2015) [119] USA	1st T. Studied from first prenatal visit (*n* = 372)	RCT.	Randomly assigned to receive: (1) daily MVM supplement, (2) “standard of care” or (3) MVM-supplement providing 750 mg choline/day.	Information processing skills	Choline supplementation +routine MVM supplements resulted in a more significant difference in visual habituation, indicating a beneficial impact on learning mechanisms involved in encoding/memory in alcohol-exposed and non/low alcohol-exposed pregnancies. This process may be mediated by the breakdown of choline to betaine and then to DMG.
Boeke et al. (2013) [120] USA	1st/2nd T. *n* = 895	Prospective cohort	Maternal choline intakes observed.	Offspring visual memory	Mean choline intake in the 2nd trimester was 328 mg and associated with modestly better child visual memory at age 7 years.
Ross et al. (2013) [121] USA	3rd T (from 17 weeks) (*n* = 100)	DB PC trial.	Received 900 mg choline/day. After birth, infants received 100 mg of phosphatidylcholine in an oral suspension once daily or placebo.	Electroencephalographic recordings	More choline-treated infants (76%) suppressed the P50 response, compared to placebo-treated infants (43%) at the fifth postnatal week (effect size 0.7). A CHRNA7 genotype associated with schizophrenia diminished P50 inhibition in the placebo-treated infants, but not in the choline-treated infants.
Cheatham et al. (2012) [122] USA	Pregnancy and postpartum (*n* = 140). Infants assessed up to 13 months.	DB RCT.	Receive supplemental phosphatidylcholine (750 mg) or a placebo (corn oil) from 18 week gestation through 90 day postpartum	Infant cognitive function	Phosphatidylcholine supplementation of pregnant women eating diets containing moderate amounts of choline did not enhance their infants’ brain function. It is possible that a longer follow-up period would reveal late-emerging effects
Villamor et al. (2012) [123] USA	1st/2nd T. *n* = 1210	Prospective cohort	Maternal choline intakes observed.	Child cognition	No associations observed between choline or cognitive outcomes at 3 years.
Wu et al. (2012) [124] Canada	2nd T. *n* = 154 mother-infant pairs.	Prospective cohort	Measured maternal plasma free choline.	Early cognitive development	Significant positive associations were found between infant cognitive test scores at 18 months of age and maternal plasma free choline at 16 weeks of gestation.
Shaw et al. (2009) [125] USA	2nd T. *n* = 180,000 pregnant women	Prospective cohort.	Serum total choline concentrations measured between 15–18 weeks gestation.	Neural tube defects	NTD risk was elevated with lower levels of total choline and reduced with higher levels of choline.
Signore et al. (2008) [126] USA	2nd/3rd T. *n* = 404 maternal-child pairs	Prospective cohort.	Serum concentrations of total and free choline measured at 16–18 week, 24–26 week, 30–32 week, and 36-38 week and in cord blood.	Intelligence	Gestational and newborn choline concentrations in the physiologic range showed no associations with childhood intelligence at 5 years.
Shaw et al. (2004) [127] USA	3 months before conception	Case -control.	Maternal dietary choline intake recorded 3 months prior to conception	Neural tube defects	NTD risk estimates were lowest amongst women whose diets were rich in choline.

Key: CHRNA7, Cholinergic Receptor Nicotinic Alpha 7 Subunit; DB, double-blind; DMG, dimethylglycine; MVM, multivitamin and mineral; NTD, neural tube defects; P50, an event-related potential occurring 50 ms after a stimulus; PC, placebo-controlled; RCT, randomized controlled trial; T, trimester.

**Table 3 nutrients-12-01731-t003:** Jadad criteria used to assess the quality of RCTs.

Publication	Randomized	Method of Randomization Described and Appropriate	Blinding Mentioned	Method of Blinding	Withdrawal and Dropout of Subjects	Total Score
Bahnfleth et al. (2019) [115]	1	*	1	*	1	3
Andrew et al. (2018) [116]	1	1	1	1	1	5
Caudill et al. (2018) [117]	1	1	1	1	1	5
Jacobson et al. (2018) [129]	1	1	1	1	1	5
Ross et al. (2016) [118]	0	0	1	1	1	3
Coles et al. (2015) [130]	1	0	0	0	0	1
Kable et al. (2015) [119]	1	1	0	0	1	3
Ross et al. (2013) [121]	1	0	1	0	1	3
Cheatham et al. (2012) [122]	1	1	1	1	1	5

Total quality assessment score for which scores range between 1 and 5: with 1 being the lowest quality and 5 being the highest quality. 3 = above average quality; * Described elsewhere.

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
