# Peer review of "Choline, Neurological Development and Brain Function: A Systematic Review Focusing on the First 1000 Days"

_nutrients, 2020, doi:10.3390/nu12061731_

Round 1
Reviewer 1 Report
This is a very in-depth and well done review on the recent science behind choline during the first 1000 days. I have only minor comments. Good work!
Abstract:
Line 20-21: I believe the authors meant to say that 813 publications were subject to title/abstract review (for maybe full-text review?), as I doubt complete information was extracted from this many articles.
Introduction:
Line 35-36: While this statement is correct... I'm not sure it is fully correct. Estrogen levels, which increase up to 60X during pregnancy drive de novo synthesis of choline. In women with SNPs in the gene PEMT this is a big issue... but in many women without these SNPs, it may not be an issue. There is not a lot of human data in this area but I suggest the authors review the articles by Ganz et al., 2017 (doi:10.3390/nu9080837) and Wallace et al. 2018 (10.1080/19390211.2019.1639875) just to see how they layout this issue. Since you are talking about betaine in the next sentence, it might also be nice to talk about how betaine in spinach, wheat, quinoa, etc. can spare some (but not all) of choline's action. This might be very important for pregnant women who are vegetarian/vegan.
Line 42: Add that the impairments resulting from this deficit can not be restored with choline repletion.
Line 44: Some forms of choline are also fat-soluble.
Line 48: metabolitE (no "s")
Line 79: Again, I'm not picky where in the introduction it is... but I would talk about genetic polymorphisms that could influence the choline requirement of a pregnant woman.
Line 108: The National Academy of Medicine in the U.S. derived values very similarly. Check the Wallace et al. 2019 review mentioned above, as he describes how recommendations were made for the US and Canada very clearly (and similar to Europe). Note the difference in deriving the AI for infants (levels in breast milk) vs. adults (liver dysfunction). It also might be worth mentioning that revision of reference intakes to include an EAR so that inadequacy and deficiency can be accurately determined would greatly advance the field. The AI is kind of just a nice guess but isn't associated with any deficiency disorder.
Line 129-133: Just double-check numbers... not saying they are wrong... USDA just uses much lower values (e.g., an egg contains 149 mg of choline)... those values in the USDA database for the choline content of foods are the ones used to calculate the percent of the population that falls below the AI in the US (and I think Europe too).
Materials and Methods:
Please update the literature search to June 1,, 2020.
Line 231-232: I'm not clear about excluding studies if PRIMA standards were not included in the publication but instead covered elsewhere... The PRISMA statement is just how to report a systematic review. Please delete or expand the explanation.
Line 240: Please explain why you limited to the last 20-years for human studies and 10-years for animal studies. Much of the depletion-repletion work on choline was conducted in the early 1990's. I think it would be smart to not put a date limit on the human studies... and then maybe work in how the animal studies within the last 10-years provide the biological mechanisms to support what human studies show.
Results:
Under the brain development and cognition and memory sections... it would be nice for the authors to review the article by Blusztajn et al. 2017 (doi:10.3390/nu9080815) as he explains this from a mechanistic standpoint very nicely. No issues with the sections... but you might wish to incorporate some of his findings as background to support the studies from the most recent 10-years.
Section 3.2.1: If you are including studies from the developing world, it would be nice to include the Lulun Study and Mazira Study. Lulun correlated the effect seen on growth and stunting to blood choline levels (Am J Clin Nutr 2017;106:1482–9)...
Discussion:
Impeccably written. My only suggestion would be to again stress the importance of choline among individuals with SNPs in folate metabolizing genes (MTHFR, etc) as choline may be of greatest importance in regard to issues like NTDs in this population.
Overall a great article.
Author Response
Reviewer report 1
This is a very in-depth and well done review on the recent science behind choline during the first 1000 days. I have only minor comments. Good work!
Abstract:
Line 20-21: I believe the authors meant to say that 813 publications were subject to title/abstract review (for maybe full-text review?), as I doubt complete information was extracted from this many articles. Many thanks this has been amended.
Introduction:
Line 35-36: While this statement is correct... I'm not sure it is fully correct. Estrogen levels, which increase up to 60X during pregnancy drive de novo synthesis of choline. In women with SNPs in the gene PEMT this is a big issue... but in many women without these SNPs, it may not be an issue. There is not a lot of human data in this area but I suggest the authors review the articles by Ganz et al., 2017 (doi:10.3390/nu9080837) and Wallace et al. 2018 (10.1080/19390211.2019.1639875) just to see how they layout this issue. This has been rewritten after reviewing the Wallace reference. Since you are talking about betaine in the next sentence, it might also be nice to talk about how betaine in spinach, wheat, quinoa, etc. can spare some (but not all) of choline's action. This might be very important for pregnant women who are vegetarian/vegan. Thank you this has additionally been added.
Line 42: Add that the impairments resulting from this deficit can not be restored with choline repletion. This has been added.
Line 44: Some forms of choline are also fat-soluble. The mention of water/fat-soluble has been removed and now simply states: Chemically, choline is closely associated with the B-vitamin family
Line 48: metabolitE (no "s") This has been amended.
Line 79: Again, I'm not picky where in the introduction it is... but I would talk about genetic polymorphisms that could influence the choline requirement of a pregnant woman. Many thanks this has been added – Ganz references.
Line 108: The National Academy of Medicine in the U.S. derived values very similarly. Check the Wallace et al. 2019 review mentioned above, as he describes how recommendations were made for the US and Canada very clearly (and similar to Europe). Note the difference in deriving the AI for infants (levels in breast milk) vs. adults (liver dysfunction). It also might be worth mentioning that revision of reference intakes to include an EAR so that inadequacy and deficiency can be accurately determined would greatly advance the field. The AI is kind of just a nice guess but isn't associated with any deficiency disorder. This has been added also thank you.
Line 129-133: Just double-check numbers... not saying they are wrong... USDA just uses much lower values (e.g., an egg contains 149 mg of choline)... those values in the USDA database for the choline content of foods are the ones used to calculate the percent of the population that falls below the AI in the US (and I think Europe too). Many thanks I have checked these from the references: https://pubmed.ncbi.nlm.nih.gov/12730414/
https://www.ncbi.nlm.nih.gov/pmc/articles/PMC6213596/ and they are aligned so will use these if you don’t mind. The USDA has rather a lot of options per food which could possibly lead to the variability. Pr Zeisel published the first set of values.
Materials and Methods:
Please update the literature search to June 1,, 2020. Updated.
Line 231-232: I'm not clear about excluding studies if PRIMA standards were not included in the publication but instead covered elsewhere... The PRISMA statement is just how to report a systematic review. Please delete or expand the explanation. This part has been removed.
Line 240: Please explain why you limited to the last 20-years for human studies and 10-years for animal studies. Much of the depletion-repletion work on choline was conducted in the early 1990's. I think it would be smart to not put a date limit on the human studies... and then maybe work in how the animal studies within the last 10-years provide the biological mechanisms to support what human studies show. This has been edited and this approach taken.
Results:
Under the brain development and cognition and memory sections... it would be nice for the authors to review the article by Blusztajn et al. 2017 (doi:10.3390/nu9080815) as he explains this from a mechanistic standpoint very nicely. No issues with the sections... but you might wish to incorporate some of his findings as background to support the studies from the most recent 10-years. Many thanks – the findings from this Blusztain paper are cited in the Choline, the Brain and Neurons section and just the findings from the ‘identified’ studies covered in the results– hope ok.
Section 3.2.1: If you are including studies from the developing world, it would be nice to include the Lulun Study and Mazira Study. Lulun correlated the effect seen on growth and stunting to blood choline levels (Am J Clin Nutr 2017;106:1482–9)... Thanks, unfortunately this didn’t come up in the keyword search – in the results just the ‘identified’ studies are included. The article I believe also measures plasma levels instead of intake. Many thanks.
Discussion:
Impeccably written. My only suggestion would be to again stress the importance of choline among individuals with SNPs in folate metabolizing genes (MTHFR, etc) as choline may be of greatest importance in regard to issues like NTDs in this population. Many thanks, we have added additional statements to reinforce this point on lines 638 and 688.
Overall a great article.
Reviewer 2 Report
The abstract has 263 words, where journal asks 200 in the instructions, although rules can be flexible in the summary format.
Introduction
There is too much text to carry out this section, since there are aspects that are repeated, others may be included within some of the sections. Within this section the fundamental aspects of choline regarding brain development are found very late. Therefore, it would be better to synthesize everything so that it would be much clearer. In addition to adding the objectives of the review at the end of this section.
Methods
The articles were evaluated with JADAD, so it would be marked on lines 232-233, (better than by relevance)
It is not specified the difference in years between the search in humans and in animals.
Why was it only searched Medline / Pubmed?
There is a language bias when searching only in English and is specified by nowhere in the text.
Table 1, when giving results, must be located in results.
Check the JADAD score of Bahnfleth et al., since it is not well scored
Why only JADAD in human articles?
Risk of bias has not been added.
Results
It is marked that in phase I, it is done in humans but in results, the first thing to appear is in animals. The sections would have to be modified for consistency.
In tables 2 and 3, in the titles of the first column put "country".
Discussion
I like this section
Conclusion
In point 3, highlight not only in development, but also in memory (line 627)
Author Response
Reviewer 2
The abstract has 263 words, where journal asks 200 in the instructions, although rules can be flexible in the summary format. Unnecessary words have been removed to make this shorter.
Introduction
There is too much text to carry out this section, since there are aspects that are repeated, others may be included within some of the sections. Within this section the fundamental aspects of choline regarding brain development are found very late. Therefore, it would be better to synthesize everything so that it would be much clearer. In addition to adding the objectives of the review at the end of this section.
The choline and brain development section has been moved up higher – after choline functions and replica statements removed.
Methods
The articles were evaluated with JADAD, so it would be marked on lines 232-233, (better than by relevance) Amended.
It is not specified the difference in years between the search in humans and in animals. This should now be clearer.
Why was it only searched Medline / Pubmed? Other databases such as Embase require high subscription fees.
There is a language bias when searching only in English and is specified by nowhere in the text. Ok, this has been removed.
Table 1, when giving results, must be located in results. This has been relocated.
Check the JADAD score of Bahnfleth et al., since it is not well scored This has been recalculated at 3.
Why only JADAD in human articles? This cannot unfortunately be applied to animal studies.
Risk of bias has not been added. The JADAD helps to decipher whether studies are at risk of bias i.e. those with lower scores and less well designed being at higher risk of bias.
Results
It is marked that in phase I, it is done in humans but in results, the first thing to appear is in animals. The sections would have to be modified for consistency. This section has been readjusted to align – animals first and the human studies.
In tables 2 and 3, in the titles of the first column put "country". This has been added thank you.
Discussion
I like this section. Many thanks.
Conclusion
In point 3, highlight not only in development, but also in memory (line 627) This has been added thank you.
Round 2
Reviewer 2 Report
Good morning:
Thanks for all your changes.
I think this article needs more minor changes:
- There is too much text in the introduction, but thanks for starting to talk earlier about the hill.
- Line 261, I think it is better to talk about "select" than "evaluate"
- Lines 263 to 269, now, there is no mention of humans or the date of their search.
- Line 291, you talk first about humans, when in lines 263 to 269 you talk about phase I is Animals ... I think the paper should be ordered in the same order (search, results, etc.); first animals and second humans, for example.
- Risk of bias for animals -> SYRCLE
- You don't talk language bias
- Why did you look only in 10 years for animals? It is still unclear, so you can talk about the latest evidence, for example
Author Response
Good morning:
Thanks for all your changes.
I think this article needs more minor changes:
- There is too much text in the introduction, but thanks for starting to talk earlier about the hill. Subheadings have been removed from the accruement section and further reduced. Further repeats have also been removed. Please could we keep what remains in please as some readers may not be familiar with the ‘background’ on choline. For many clinicians etc. choline is a new nutrient that they are not familiar with.
- Line 261, I think it is better to talk about "select" than "evaluate" Amended thank you.
- Lines 263 to 269, now, there is no mention of humans or the date of their search. This has been put back in.
- Line 291, you talk first about humans, when in lines 263 to 269 you talk about phase I is Animals ... I think the paper should be ordered in the same order (search, results, etc.); first animals and second humans, for example. Thank you, this has been reorganised and rechecked throughout
- Risk of bias for animals -> SYRCLE Thank you, an excellent resource I had not heard of previously. We have applied this.
- You don't talk language bias This has now been added after reading the Hooijmans papers.
- Why did you look only in 10 years for animals? It is still unclear, so you can talk about the latest evidence, for example This has been further explained. Previous reviews have collated earlier evidence for which there is an extensive body of science. These have been cited.